# Trithorax maintains the functional heterogeneity of neural stem cells through the transcription factor Buttonhead

Hideyuki Komori[1†], Qi Xiao[2†], Derek H Janssens[3], Yali Dou[4], Cheng-Yu Lee[1,2,3,5]*

[1]Center for Stem Cell Biology, Life Sciences Institute, University of Michigan Medical School, Ann Arbor, United States; [2]Department of Cell and Developmental Biology, University of Michigan Medical School, Ann Arbor, United States; [3]Program in Cell and Molecular Biology, University of Michigan Medical School, Ann Arbor, United States; [4]Department of Pathology, University of Michigan Medical School, Ann Arbor, United States; [5]Division of Molecular Medicine and Genetics, Department of Internal Medicine, University of Michigan Medical School, Ann Arbor, United States

**Abstract** The mechanisms that maintain the functional heterogeneity of stem cells, which generates diverse differentiated cell types required for organogenesis, are not understood. In this study, we report that Trithorax (Trx) actively maintains the heterogeneity of neural stem cells (neuroblasts) in the developing *Drosophila* larval brain. *trx* mutant type II neuroblasts gradually adopt a type I neuroblast functional identity, losing the competence to generate intermediate neural progenitors (INPs) and directly generating differentiated cells. Trx regulates a type II neuroblast functional identity in part by maintaining chromatin in the *buttonhead* (*btd*) locus in an active state through the histone methyltransferase activity of the SET1/MLL complex. Consistently, *btd* is necessary and sufficient for eliciting a type II neuroblast functional identity. Furthermore, over-expression of *btd* restores the competence to generate INPs in *trx* mutant type II neuroblasts. Thus, Trx instructs a type II neuroblast functional identity by epigenetically promoting Btd expression, thereby maintaining neuroblast functional heterogeneity.

*For correspondence:
leecheng@umich.edu

†These authors contributed equally to this work

**Competing interests:** The authors declare that no competing interests exist.

## Introduction

Stem cells employ several strategies to generate the requisite number of diverse differentiated cell types required for organ development and organ homeostasis in higher eukaryotes (*Franco and Müller, 2013*; *Kohwi and Doe, 2013*). One such strategy involves stem cells changing their temporal identities. For example, neuroblasts sequentially express distinct temporal-identity transcription factors, allowing them to generate diverse differentiated cells in the fly embryonic ventral nerve cord (*Isshiki et al., 2001*; *Pearson and Doe, 2003*). Another strategy involves maintaining a functionally heterogeneous pool of tissue-specific stem cells. Studies in flies and vertebrate systems show that functionally heterogeneous stem cells directly contribute to the generation of diverse cell types during hematopoiesis, gut homeostasis, and brain development (*Barker et al., 2007*; *Bello et al., 2008*; *Boone and Doe, 2008*; *Bowman et al., 2008*; *Graf and Stadtfeld, 2008*; *Copley et al., 2012*; *Franco et al., 2012*; *Marianes and Spradling, 2013*). Numerous patterning mechanisms have been described to explain how the fates of distinct stem cells within a developing organ become specified, but how their functional heterogeneity is maintained throughout the lifespan of an organism remains completely unknown.

**eLife digest** Whereas the majority of cells in the brain are unable to divide to produce new cells, neural stem cells can divide numerous times and have the potential to become many different types of brain cells. However, between these two extremes there is another group of cells called neural progenitors. These cells can give rise to multiple types of neurons but, in contrast to stem cells, they can undergo only a limited number of divisions.

Many of the molecular mechanisms by which stem cells give rise to progenitors are similar in mammals and in the fruit fly *Drosophila*. In the brains of fly larvae, a subset of neural stem cells called type II neuroblasts give rise to 'intermediate neural progenitors', each of which can divide between four and six times. Every division generates a replacement intermediate neural progenitor and a cell called a ganglion mother cell, which divides one last time to produce two brain cells. Thus, intermediate neural progenitors increase the overall output of cells derived from every division of a type II neuroblast.

The ability of type II neuroblasts to generate intermediate neural progenitors is important for development. Loss of this ability will result in a shortage of cells, disrupting brain development, while the faulty generation of intermediate neural progenitors will result in the formation of tumors. Now, using *Drosophila* brain cells cultured in the laboratory, Komori et al. show that an evolutionarily conserved enzyme called Trithorax has an important role in maintaining this ability. Trithorax acts through a protein called Buttonhead. The role of Buttonhead in regulating intermediate neural progenitors has also been identified by Xie et al.

Komori et al. show that type II neuroblasts that lack Trithorax activity lose their unique identity and behave as type I neuroblasts, which never generate intermediate neural progenitors. Trithorax maintains the cellular memory of a type II neuroblast by keeping regions of chromatin—a macromolecule made of DNA and proteins called histones—in an active state. These regions contain key genes, such as the gene for Buttonhead. Re-introducing Buttonhead in type II neuroblasts that lack Trithorax activity can reinstate their ability to produce intermediate neural progenitors.

The central complex of the insect brain is comprised of an intricate network of neurons and glia that process a vast number of environmental inputs essential for daily life (**Boyan and Reichert, 2011**; **Boyan and Williams, 2011**). All differentiated cell types in the central complex arise from repeated rounds of self-renewing asymmetric divisions of type I and type II neuroblasts, which are molecularly and functionally distinct (**Bello et al., 2008**; **Boone and Doe, 2008**; **Bowman et al., 2008**) (**Figure1—figure supplement 1**). In every asymmetric division, a type I neuroblast always generates a precursor cell (ganglion mother cell or GMC) that divides once to produce two differentiated cells. By contrast, every asymmetric division of a type II neuroblast invariably leads to the generation of an immature INP that acquires an INP functional identity during maturation. An INP undergoes 5–8 rounds of asymmetric division to regenerate and generate a GMC with each division (**Homem et al., 2013**). Thus, the ability to generate INPs functionally distinguishes these two types of neuroblasts.

Type II neuroblasts uniquely express the ETS transcription factor Pointed P1 (PntP1) (**Zhu et al., 2011**; **Xiao et al., 2012**). Mis-expression of PntP1 can induce a type II neuroblast functional characteristic in a type I neuroblast (**Zhu et al., 2011**). However, the physiological function of PntP1 in the maintenance of a type II neuroblast functional identity remains unclear. The *pnt* locus encodes at least three distinct alternatively spliced transcripts. Thus, it is formally possible that multiple isoforoms of Pnt or a yet unknown mechanism function to maintain a type II neuroblast functional identity.

Epigenetic mechanisms such as the methylation of histone H3 Lysine 4 (H3K4) play central roles in specifying cell type identities during development (**Lim et al., 2009**; **Ang et al., 2011**; **Schuettengruber et al., 2011**; **Shilatifard, 2012**; **Yang et al., 2012**). The evolutionarily conserved SET1/Mixed-lineage leukemia (MLL) complexes catalyze the methylation of H3K4 and maintain the target gene loci in a transcriptionally active state (**Miller et al., 2001**; **Roguev et al., 2001**; **Krogan et al., 2002**). The fly genome encodes three orthologs of the SET1/MLL protein, Trx, Trithorax-related (Trr), and dSet1. Similar to their mammalian counterparts, Trx, Trr, or dSet1 can each assemble functionally active complexes by binding to Absent, small, or homeotic discs 2 (Ash2), Retinoblastoma binding protein 5 (Rbbp5), and will die slowly (Wds) (**Wu et al., 2008**; **Ardehali et al., 2011**; **Mohan et al., 2011**).

Functionally, Trr or dSet1 regulates global mono- or tri-methylation of H3K4 respectively. In contrast, Trx appears to selectively regulate the expression of the *Hox* genes through the methylation of H3K4 (*Breen and Harte, 1993*; *Yu et al., 1995*). However, little is known about the targets of Trx beyond the *Hox* genes.

Here, we report that Trx maintains the type II neuroblast functional identity by regulating the transcription of *btd* during fly larval brain neurogenesis. Type II neuroblasts mutant for *trx* or genes encoding the core components of the SET1/MLL complex display a type I neuroblast marker expression profile and generate GMCs instead of INPs. These results indicate that Trx maintains a type II neuroblast functional identity by regulating the transcription of specific target genes. We identified a direct downstream target of Trx, Btd, that plays an important role in the maintenance of a type II neuroblast functional identity. *btd* mutant type II neuroblasts adopt a type I neuroblast functional identity and directly generate GMCs instead of INPs. Conversely, type I neuroblasts over-expressing *btd* assume a type II neuroblast functional identity and generate INP progeny. Most importantly, over-expression of *btd* restores the competence of *trx* mutant type II neuroblasts to generate INPs. Thus, we conclude that Trx functions to epigenetically maintain Btd expression in type II neuroblasts, thereby maintaining neuroblast functional heterogeneity in the larval brain.

## Results

### *trx* regulates neuroblast heterogeneity by maintaining a type II neuroblast identity

Analyses of gene transcription in mutant larval brains enriched with type I or type II neuroblasts led us to hypothesize that differential regulation of gene expression contributes to neuroblast functional heterogeneity (*Carney et al., 2012*) (Komori and Lee, unpublished observation). Because the *trx* gene contributes to cell fate maintenance in a variety of developmental processes, we tested whether it is required for maintaining neuroblast heterogeneity. We induced GFP-marked mosaic clones derived from single wild-type or *trx* mutant type I or II neuroblasts and assessed the identities of cells in the clones by examining the expression of cell fate markers in a time-course study (*Figure 1—figure supplement 1*). Identical to wild-type neuroblasts, *trx* mutant type I neuroblasts maintained the expression of Deadpan (Dpn) and Asense (Ase) and the cytoplasmic localization of Prospero (Pros), but lacked PntP1 expression (Dpn$^+$Ase$^+$PntP1$^-$Pros$^{cytoplasmic}$) (*Table 1*, data not presented). In addition, both wild-type and *trx* mutant type I neuroblasts were always surrounded by GMCs (Dpn$^-$Ase$^+$Pros$^{nuclear}$) (data not presented). Thus, Trx is dispensable for the maintenance of a type I neuroblast functional identity. While all wild-type type II neuroblasts displayed a Dpn$^+$Ase$^-$PntP1$^+$Pros$^-$ marker expression profile in all stages examined, *trx* mutant type II neuroblasts progressively altered their marker expression profile (*Figure 1A–D*, *Table 1*). Strikingly, almost all *trx* mutant type II neuroblasts in 72-hr clones displayed a type I neuroblast marker expression profile (*Figure 1B–D*; *Table 1*). These data strongly suggest that *trx* mutant type II neuroblasts adopt a type I neuroblast identity.

We extended our analyses to examine the identity of progeny directly derived from *trx* mutant type II neuroblasts. We observed a time-dependent reduction in INPs in *trx* mutant type II neuroblast clones as compared to identically staged wild-type clones. At 72 hr after clone induction, a control type II neuroblast was surrounded by approximately 20 INPs and 12 INP-derived GMCs that can be unambiguously identified by the expression of an *erm*-lacZ reporter transgene (*Figure 1C,E–F,H*, *Figure 1—figure supplement 1*). In contrast, an identically staged *trx* mutant neuroblast was directly surrounded by non-neuroblast progeny that displayed a Dpn$^-$Ase$^+$Pros$^{nuclear}$*erm*-lacZ$^-$ expression

**Table 1.** Summary of the marker expression profile in various genetic backgrounds

| Genotype | Neuroblast type | Dpn | Ase | Pros* | PntP1 |
|---|---|---|---|---|---|
| wild-type | I | + | + | + | − |
| wild-type | II | + | − | − | + |
| *Trx*$^{-/-}$ | I | + | + | + | − |
| *Trx*$^{-/-}$ | II | + | + | + | − |
| *Rbbp5*$^{-/-}$ | I | + | + | + | − |
| *Rbbp5*$^{-/-}$ | II | + | + | + | − |
| *btd*$^{-/-}$ | I | + | + | + | − |
| *btd*$^{-/-}$ | II | + | − | − | + |

'+' indicates detected marker expression whereas '−' indicates lack of marker expression. '*' indicates basal asymmetric localization at the basal cortex in mitotic neuroblasts.

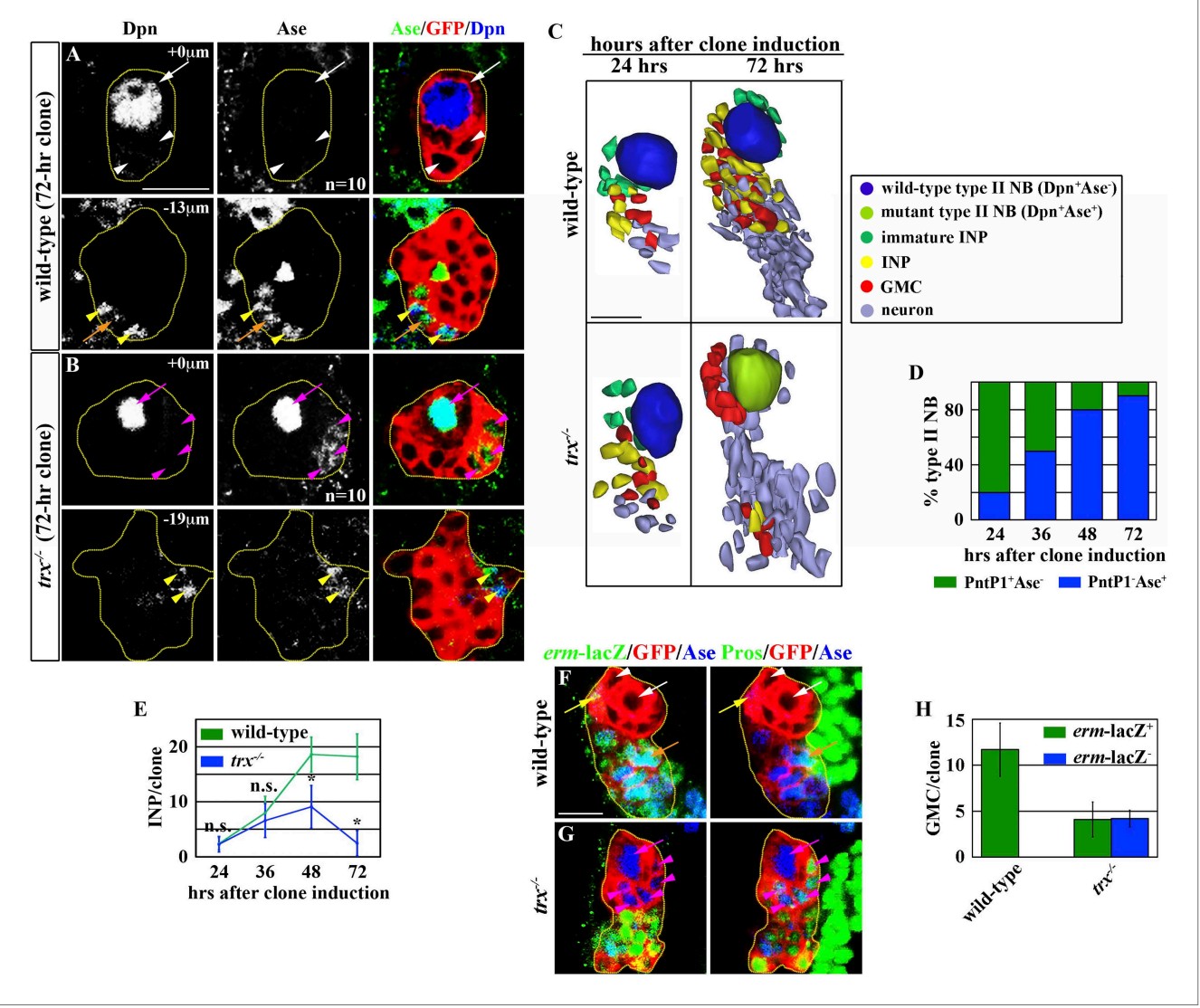

**Figure 1**. *trx* mutant type II neuroblasts display characteristics of a type I neuroblast. Key for all figures: all clones are outlined in yellow. Wild-type type II neuroblasts or mutant type I neuroblasts (Dpn+Ase−Pros−; white arrow); Ase− immature INPs (Dpn−Ase−Pros−; white arrowhead); Ase+ immature INPs (Dpn−Ase+Pros−; yellow arrow); INPs (Dpn+Ase+*erm*-lacZ+Pros^cytoplasmic; yellow arrowhead); GMC generated by INPs (Ase+*erm*-lacZ+Pros^nuclear; orange arrow); wild-type type I neuroblasts or mutant type II neuroblasts (Dpn+Ase+Pros^cytoplasmic; magenta arrow); GMC generated by wild-type type I neuroblasts or mutant type II neuroblasts (Ase+Pros+*erm*-lacZ−; magenta arrowhead). Single asterisks indicate a statistically significant (p-value <0.05) difference between the marked genotype and the control genotype in the same bar graph, as determined by the Student's t-test. n.s. indicates that the difference is statistically insignificant. NB: neuroblast. (**A**–**D**) *trx* mutant type II neuroblasts progressively acquire a type I neuroblast identity. (**A**–**B**) In the 72-hr GFP-marked clone, a wild-type type II neuroblast displays a Dpn+Ase− marker expression profile whereas a *trx* mutant type II neuroblast displays a Dpn+Ase+ expression profile. Scale bar, 10 μm. (**C**) Three-dimensionally reconstructed images of type II neuroblasts clones of the indicated genotypes. Scale bar, 10 μm. (**D**) The frequency of *trx* mutant type II neuroblasts displaying a type I neuroblast maker expression profile (PntP1-Ase−). N = 10 per time point. (**E**–**H**) *trx* mutant type II neuroblasts lose the ability to generate INPs. (**E**) The average number of INPs per staged type II neuroblast clone of the indicated genotype. N = 10 per time point. (**F**–**G**) In the 72-hr GFP-marked clones, a wild-type type II neuroblast is surrounded by INPs and their GMC progeny identified by *erm*-lacZ expression. In contrast, a *trx* mutant type II neuroblast is surrounded by GMCs that are directly derived from neuroblasts and lack *erm*-lacZ expression. Scale bar, 10 μm. (**H**) The average number of GMCs with or without *erm*-lacZ expression per type II neuroblast clone of the indicated genotypes.

The following figure supplement is available for figure 1:

**Figure supplement 1**. A diagram of two distinct neuroblast lineages.

profile identical to GMCs derived from type I neuroblasts (*Figure 1C,G*, *Figure 1—figure supplement 1*). Although *trx* mutant clones also contained an average of 3 INPs and 4 INP-derived GMCs, these cells were located at the extreme distal end of the clone, consistent with *trx* mutant type II neuroblasts adopting a type I neuroblast identity following the clone induction (*Figure 1C,E,G–H*). These data strongly suggest that Trx regulates neuroblast heterogeneity by maintaining a type II neuroblast identity.

### *trx* maintains the functional identity of type II neuroblasts

The competence to generate INPs is a main feature that distinguishes the functional identity of a type II neuroblast from that of a type I neuroblast (*Weng and Lee, 2011*; *Homem and Knoblich, 2012*; *Janssens and Lee, 2014*). *brain tumor* (*brat*) and *erm* function in the immature INP to promote INP identity specification in the type II neuroblast lineage, and the defective specification of an INP identity leads to the formation of supernumerary type II neuroblasts in the *brat* or *erm* mutant brain (*Xiao et al., 2012*; *Eroglu et al., 2014*; *Janssens et al., 2014*; *Koe et al., 2014*; *Komori et al., 2014*). If *trx* mutant type II neuroblasts indeed adopt a type I neuroblast functional identity, their progeny should be insensitive to the loss of *brat* or *erm* function and generate differentiated cells instead of reverting into supernumerary neuroblasts. A control type II neuroblast clone in the *brat* mutant brain contained more than 100 supernumerary type II neuroblasts and was devoid of GMCs and neurons (*Figure 2A,E*). By contrast, a *trx* mutant type II neuroblast clone in the *brat* mutant brain contained far fewer supernumerary type II neuroblasts and far more GMCs and neurons as compared to the control clone (*Figure 2A–B,E*). Similarly, a control type II neuroblast clone in the *erm* mutant brain contained more than 50 supernumerary type II neuroblasts and few GMCs and neurons (*Figure 2C,E*). In contrast, a *trx* mutant type II neuroblast clone in the *erm* mutant brain contained fewer supernumerary type II neuroblasts but more GMCs and neurons as compared to the control clone (*Figure 2C–E*). Together, these data strongly suggest that *trx* mutant type II neuroblasts lost the competence to generate immature INPs.

We directly tested whether *trx* mutant type II neuroblasts adopt a type I neuroblast functional identity and directly generate GMCs. Pros segregates exclusively into GMCs where it suppresses a type I neuroblast functional identity during asymmetric division of a type I neuroblast, but is undetectable in mitotic type II neuroblasts (*Knoblich et al., 1995*; *Spana and Doe, 1995*; *Choksi et al., 2006*; *Bayraktar et al., 2010*). In a telophase *trx* mutant type II neuroblast, however, Pros localized asymmetrically in the basal cortex and segregated uniquely into the cortex of the future non-neuroblast progeny (*Figure 2F–H*). Most importantly, removing *pros* function in *trx* mutant type II neuroblasts leads to the formation of supernumerary type I neuroblasts (*Figure 2I*). These data confirm that *trx* mutant type II neuroblasts adopt a type I neuroblast functional identity and directly generate GMCs. Thus, we conclude that *trx* regulates neuroblast heterogeneity by maintaining a type II neuroblast functional identity.

### Trx maintains the type II neuroblast functional identity through the histone methyltransferase activity of the SET1/MLL complex

We assessed whether the histone methylation activity of Trx is required for maintaining a type II neuroblast functional identity. We induced mosaic clones derived from type II neuroblasts carrying the *trx*$^{Z11}$ allele, which results in a missense mutation in the SET domain of Trx and reduces the histone methyltransferase activity of the Trx protein (*Smith et al., 2004*; *Tie et al., 2014*). Twenty-seven percent of *trx*$^{Z11}$ type II neuroblasts assumed a type I neuroblast functional identity as determined by both the expression of a type I neuroblast marker expression profile and the generation of GMCs (*Figure 3A–B*). This result indicates that the histone methylation activity of Trx is essential for the maintenance of a type II neuroblast functional identity. Trx was co-purified with the core components of the SET1/MLL complex, Ash2, Rbbp5, and Wds, from the lysate extracted from S2 cells (*Mohan et al., 2011*). Thus, we tested whether the core components of the SET1/MLL complex are required for maintaining a type II neuroblast identity. Indeed, knocking down the function of *ash2*, *rbbp5*, or *wds* individually leads to fewer type II neuroblasts and INPs per brain lobe, identical to reducing *trx* function (*Figure 3—figure supplement 1A–G*). Together, these data strongly support our hypothesis that Trx maintains a type II neuroblast functional identity through the SET1/MLL complex via a mechanism dependent of the histone methyltransferase activity.

We focus on the Rbbp5 protein, which is essential for eliciting the histone methyltransferase activity of the SET1/MLL complex (*Cao et al., 2010*), to test whether Trx maintains a type II neuroblast functional identity through the SET1/MLL complex. We first generated a null allele of the *rbbp5* gene

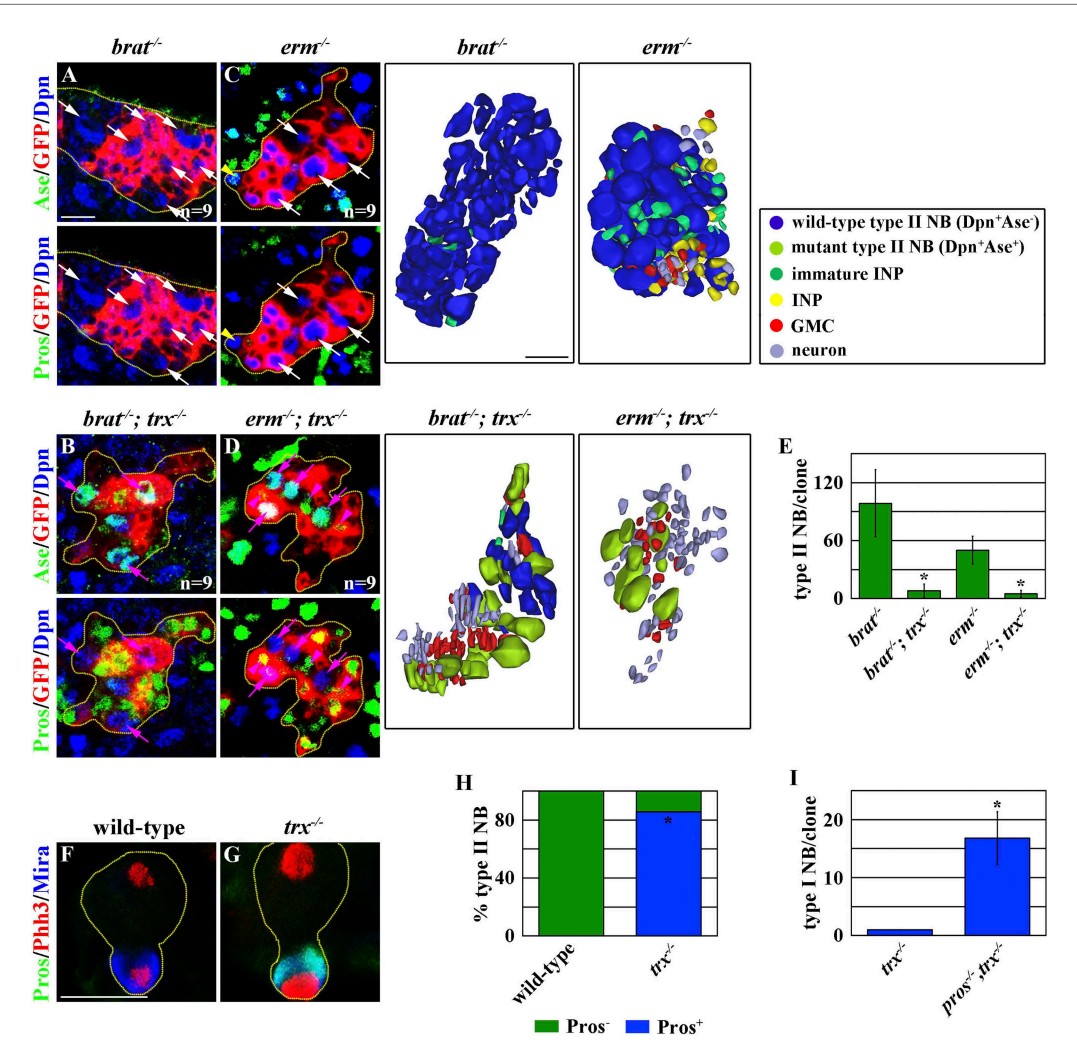

**Figure 2**. *trx* mutant type II neuroblast directly generates GMCs. (**A–E**) *trx* is required for the expansion of supernumerary type II neuroblasts in the *brat* or *erm* mutant. (**A–D**) Removing *trx* function suppresses the expansion of supernumerary type II neuroblasts and restores differentiation in the 96-hr *brat* or *erm* mutant type II neuroblast clones. Three-dimensionally reconstructed images of the clones are shown to the right. Scale bar, 10 μm. (**E**) The average number of type II neuroblasts per clone of the indicated genotypes. (**F–I**) *trx* mutant type II neuroblasts exclusively distribute Pros to their progenies to specify GMC identity. (**F–G**) In the 48-hr clones, a wild-type type II neuroblast shows undetectable expression of Pros in telophase, whereas a *trx* mutant type II neuroblast shows the basal cortical localization of Pros. Scale bar, 10 μm. (**H**) The frequency of wild-type or *trx* mutant mitotic type II neuroblasts displaying the basal localization of Pros. (**I**) The average number of type I neuroblasts per type II neuroblast clone of the indicated genotypes at 72 hr after clone induction.

(*rbbp5^null*) by excising a transposable P-element inserted at the 5′ end from the transcription start site (**Figure 3—figure supplement 2A**). Mutant analyses confirmed that *rbbp5^null* type II neuroblasts indeed adopt a type I neuroblast functional identity (**Figure 3C–F**, **Table 1**, **Figure 3—figure supplement 2B**). Thus, a *rbbp5^null* type II neuroblast is phenotypically indistinguishable from a *trx* mutant type II neuroblast. We next examined the H3K4 methylation pattern in the *rbbp5^null* type II neuroblast. All cells in the clones derived from single *rbbp5^null* type II neuroblast showed undetectable mono- and tri-methylation of H3K4 (**Figure 3G**, data not presented). This result is consistent with the SET1/MLL complex exerting its regulatory functions through the H3K4 methylation. Most importantly, over-expression of a *UAS-rbbp5^FL* transgene that encodes a full-length Rbbp5 completely restored a type II neuroblast functional identity and significantly restored both the H3K4 mono- and tri-methylation in

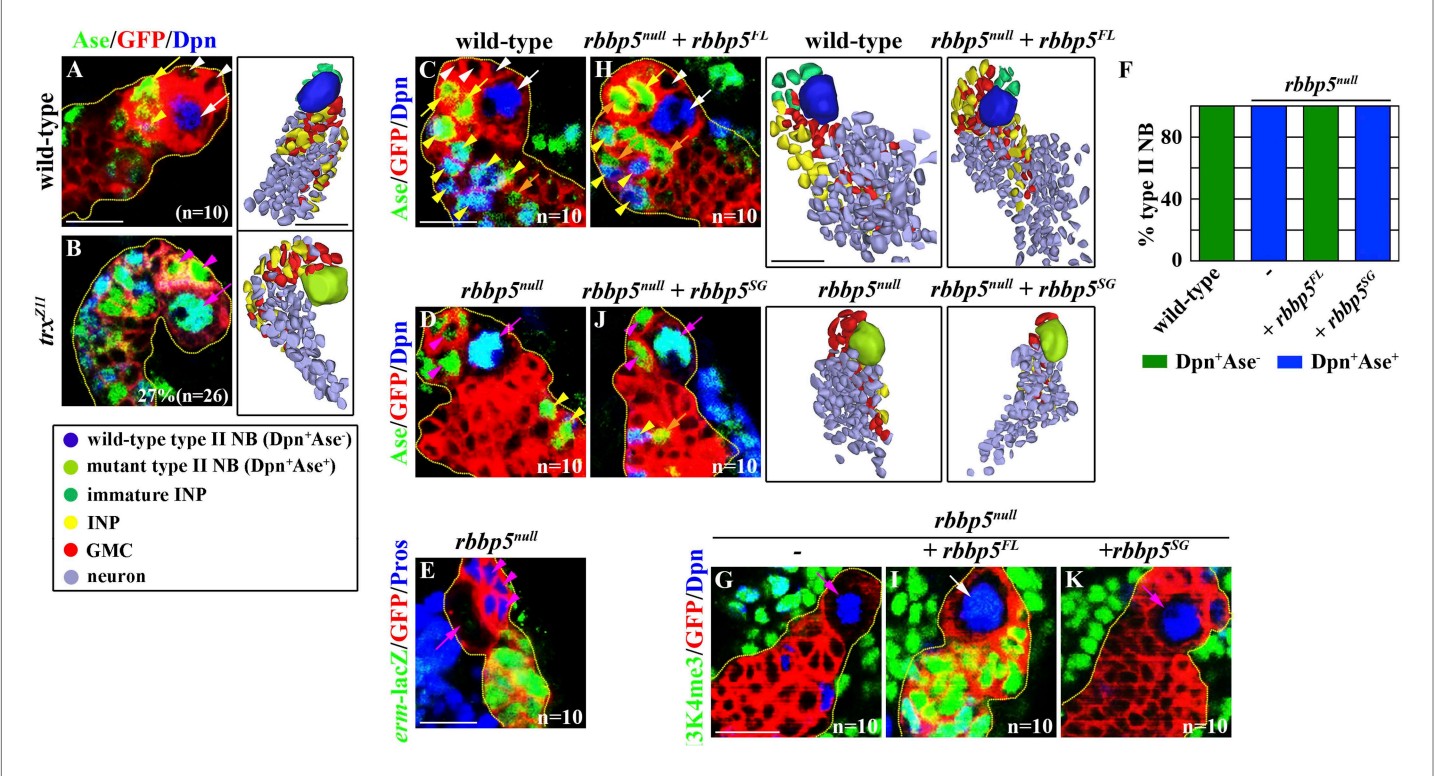

**Figure 3**. Trx and the core components of the SET/MLL complex maintain a type II neurobalst functional identity dependently on their catalytic activity for H3K4 methylaiton. (**A–B**) The function of *trx* for the H3K4 methylation is required for the maintenance of a type II neuroblast functional identity. (**A–B**) In the 72-hr clones, a *trx^Z11* mutant type II neuroblast displays a type I neuroblast marker expression profile and directly generates GMCs. Scale bar, 10 μm. Three-dimensionally reconstructed images of the clones are shown to the right. Scale bar, 10 μm. (**C–K**) The function of *rbbp5* for the H3K4 methylation is required for the maintenance of a type II neuroblast functional identity. (**C–E, H, J**) In the 96-hr clones, *rbbp5^null* type II neuroblasts display a type I neuroblast marker expression profile and directly generate GMCs. Over-expression of *rbbp5^FL* but not *rbbp5^SG* restores a type II neuroblast functional identity in *rbbp5^null* type II neuroblasts. Three-dimensionally reconstructed images of the clones are shown to the right. Scale bar, 10 μm. (**F**) The frequency of type II neuroblasts of the indicated genotypes displaying the type I or type II marker expression profiles. (**G, I, K**) *rbbp5* function is essential for the H3K4 methylation in fly larval brains. Scale bar, 10 μm.

The following figure supplements are available for figure 3:

**Figure supplement 1**. Decreasing the function of the core components of the SET1/MLL complex leads to a reduction in type II neuroblasts.

**Figure supplement 2**. Generation of the *rbbp5^null* allele and the *UAS-rbbp5^SG* transgene.

*rbbp5^null* type II neuroblasts (***Figure 3F,H–I***, ***Figure 3—figure supplement 2B***, data not presented). By contrast, over-expression of a *UAS-rbbp5^SG* transgene, which encodes a mutant Rbbp5 protein predicted to perturb the histone methyltransferase activity of the SET1/MLL complex (***Figure 3—figure supplement 2C***) (***Cao et al., 2010***), failed to restore a type II neuroblast functional identity and the methylation of H3K4 in *rbbp5^null* type II neuroblasts (***Figure 3F,J–K***, ***Figure 3—figure supplement 2B***, data not presented). Similarly, type II neuroblasts bearing a strong *ash2* mutant allele also adopted a type I neuroblast functional identity and lost most H3K4 methylation based on the same criteria (data not presented). Thus, the histone methyltransferase activity of the SET1/MLL complex is required for the maintenance of a type II neuroblast identity. We conclude that Trx maintains a functional identity of type II neuroblasts through the histone methylation activity of the SET1/MLL complex.

## Trx regulates a type II neuroblast functional identity by maintaining an active chromatin state in the *btd* locus

Knocking down the function of *trr* or *dset1* drastically reduced the global H3K4 mono- or tri-methylation in type II neuroblasts but had no effects on the maintenance of their functional identity

(*Figure 4—figure supplement 1A–J*). By contrast, removing *trx* function had no appreciable effects on the global H3K4 pattern in type II neuroblasts (*Figure 4—figure supplement 1K–N*). These data led us to hypothesize that Trx maintains the type II neuroblast functional identity by regulating a small number of genes that are specifically expressed in the type II neuroblast. We compared gene transcription profiles by using mRNAs isolated from dissected larval brains enriched with type I or II neuroblasts to identify the candidate Trx target genes (*Bowman et al., 2008*; *Weng et al., 2010*; *Carney et al., 2012*; *Haenfler et al., 2012*). *pnt* and *btd* were among a small number of genes that were dramatically up-regulated in the mRNAs isolated from larval brains enriched with type II neuroblasts as compared to the mRNAs isolated from larval brains enriched with type I neuroblasts. We confirmed that both *pntP1* and *btd* transcripts were indeed highly enriched in the brain lysate enriched with type II neuroblasts by qRT-PCR (*Figure 4A*). Furthermore, we detected the binding of Trx to the transcription start site for both the *pntP1* and *btd* transcription units (*Figure 4B*, *Figure 4—figure supplement 2A*). In addition, the promoter region of both the *pntP1* and *btd* transcription units also displayed a high level of H3K4 di-methylation, consistent with Trx-maintaining chromatin in an active state in these two loci through the H3K4 methylation (*Figure 4B*, *Figure 4—figure supplement 2A*). By contrast, we did not detect Trx binding to the negative control region located 7.5 kilobases 3′ from the *btd* transcription unit (*Figure 4B*; data not presented) (*Petruk et al., 2012*). Thus, both *pnt* and *btd* are the direct target genes of Trx.

We next tested whether either one of these two genes might regulate a functional identity of type II neuroblasts.

1. *pnt*: because the *pnt* locus encodes multiple alternatively spliced transcripts, we assessed the function of *pnt* in the type II neuroblast by over-expressing three independent *UAS-RNAi* transgenes targeting two different regions of the same exon shared by all *pnt* transcripts (*Figure 4—figure supplement 2A*). All three RNAi transgenes efficiently reduced *pnt* expression as indicated by a drastic reduction in the PntP1 protein (*Figure 4—figure supplement 2B–C*; data not presented). Unexpectedly, knocking down the function of *pnt* in type II neuroblasts led to the formation of supernumerary neuroblasts (*Figure 4—figure supplement 2D–F*). These results strongly suggest that *pnt* functions in the immature INP to promote INP identity specification similar to *brat* and *erm*. Consistently, heterozygosity of the *pnt* locus strongly enhanced the supernumerary neuroblast phenotype in the *brat* or *erm* hypomorphic brain (*Figure 4—figure supplement 2G*). In addition, overexpression of *pntP1* failed to restore a type II neuroblast functional identity in *trx* mutant type II neuroblasts (data not presented). Thus, we conclude that *pnt* functions downstream of *trx* to specify an INP identity in the immature INP rather than to maintain the type II neuroblast functional identity.

2. *btd*: a specific antibody against Btd is currently unavailable, and a genomic transgene that carries a BAC clone containing the entire *btd* locus led to embryonic lethality (Komori and Lee, unpublished). Thus, we determined the spatial expression pattern of the *btd* gene by examining the expression of a *btd-Gal4* transgene containing an enhancer element that was bound by Trx and displayed a high level of the di-methylation of H3K4 located 5 Kb upstream from the *btd* transcription start site (*Figure 4B*). The expression of a *UAS* reporter transgene driven by *btd-Gal4* was detected specifically in type II neuroblasts but was undetectable in type I neuroblasts in wild-type brains (*Figure 4C*). Importantly, the expression of *btd-Gal4* was drastically reduced in *rbbp5^null^* mutant brains (*Figure 4D*). Together, these data strongly support our hypothesis that *btd* is an excellent candidate for functioning downstream of *trx* to maintain the type II neuroblast functional identity.

If Trx maintains a type II neuroblast functional identity by regulating *btd* transcription, removing *btd* function should trigger type II neuroblasts to adopt a type I neuroblast functional identity. We assessed the identities of cells in the clones derived from single *btd* mutant type II neuroblasts by examining cell fate marker expression. *btd* mutant type II neuroblasts maintained a type II neuroblast marker expression profile in all stages examined, but these clones displayed a time-dependent reduction in INPs (*Figure 4F–G*). Unlike the control clone, however, INPs in the 72-hr *btd* mutant clone were always located at the extreme distal end of the clone (data not presented). In these clones, *btd* mutant type II neuroblasts were surrounded by 1–2 progeny resembling Ase⁻ immature INPs but never Ase⁺ immature INPs (*Figure 4F*). Instead, the remaining cells directly adjacent to the *btd* mutant type II neuroblast displayed a marker expression profile indicative of GMCs and immature neurons that are

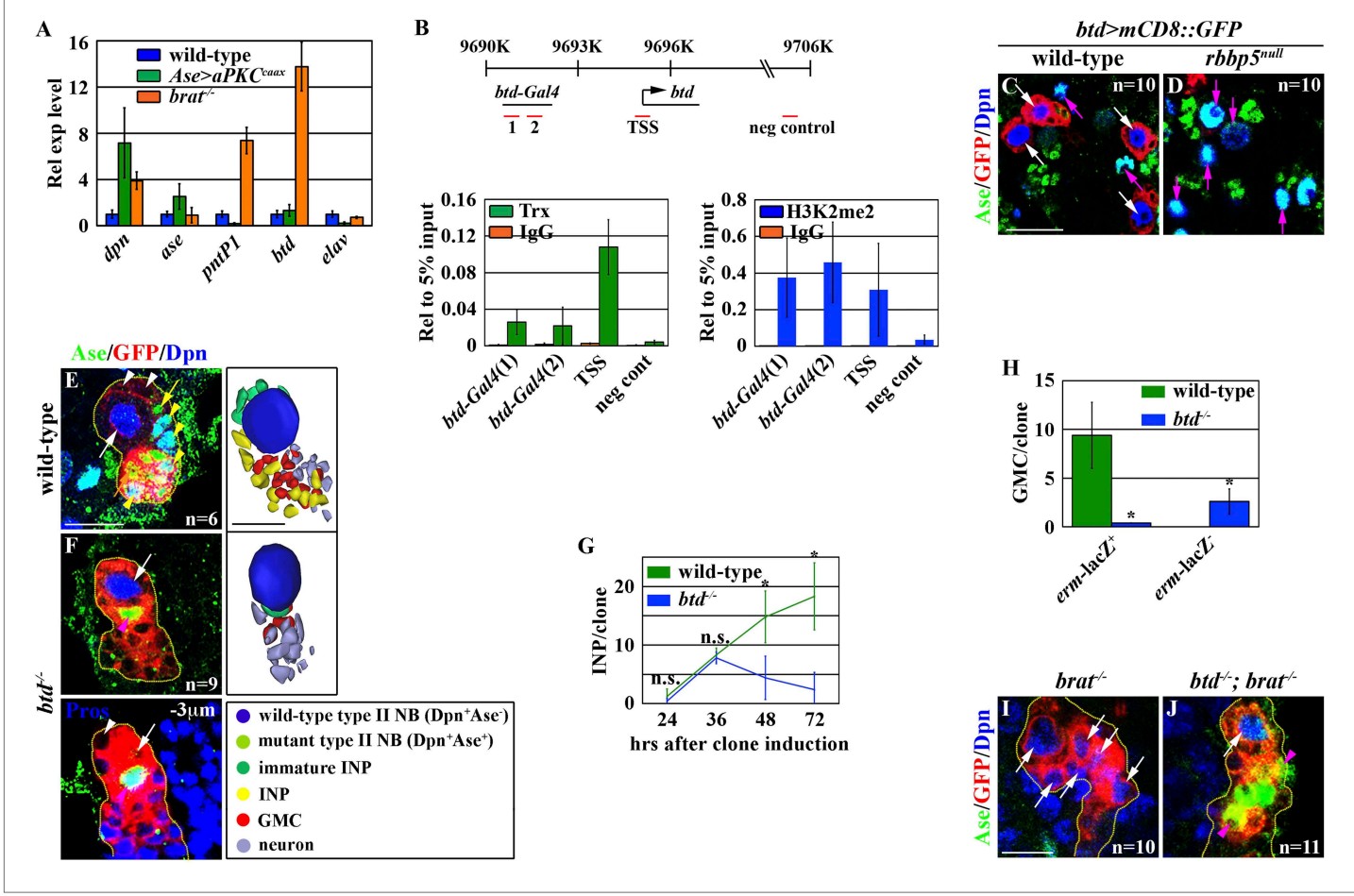

**Figure 4**. Btd likely acts downstream of Trx to maintain a type II neuroblast functional identity. (**A–D**) The *btd* gene is an excellent candidate target of Trx in the type II neuroblast. (**A**) The *btd* mRNA is highly enriched in the lysate extracted from larval brain enriched with type II neuroblasts. The *elav* transcript is highly enriched in differentiated neurons. The quantification represents the average of three biological replicates. (**B**) Trx directly binds to the type II neuroblast-specific enhancer element as well as the transcription start site (TSS) of the *btd* gene. The ChIP experiments were performed using the extract isolated from dissected *brat* mutant brains that are enriched with type II neuroblasts. Quantification of chromatin immunoprecipitated by the indicated antibodies relative to 5% of input. The quantification represents the average of three biological replicates. (**C–D**) An enhancer element from the *btd* gene is sufficient to induce type II neuroblast-specific expression of a *UAS-mCD8::gfp* reporter transgene in wild-type brain, while the enhancer activity of btd-Gal4 was reduced in *rbbp5*null brain. Scale bar, 20 μm. (**E–H**) *btd* is required for maintaining the functional identity but not the molecular signature of a type II neuroblast. (**E–F**) In the 72-hr clones, *btd* mutant type II neuroblasts maintain a type II neuroblast marker expression profile and are surrounded by 1–2 immature INP-like cells. Three-dimensionally reconstructed images of the clones are shown below. Scale bar, 10 μm. (**G**) The average number of INPs per clone of the indicated genotypes. (**H**) The average number of GMCs with or without *erm*-lacZ expression per type II neuroblast clones of the indicated genotypes at 72 hr after clone induction. (**I–J**) The immature INP-like cells generated by *btd* mutant type II neuroblasts are insensitive to loss of *brat* function. Removing *brat* function does not lead to supernumerary neuroblast formation in the 72-hr *btd* mutant type II neuroblast clones. Scale bar, 10 μm.

The following figure supplements are available for figure 4:

**Figure supplement 1**. Global H3K4 mono- or tri-methylation is not required for maintenance of a type II neuroblast functional identity.

**Figure supplement 2**. Pnt likely functions to specify an INP identity.

normally found in the type I neuroblast lineage (*Figure 4F,H*). These observations prompted us to test whether the progeny of the *btd* mutant type II neuroblast resembling Ase⁻ immature INPs were indeed functional by examining their dependency on *brat* function. In the *brat* mutant type II neuroblast clone, Ase⁻ immature INPs rapidly reverted to supernumerary neuroblasts (*Figure 4I*) (*Xiao et al., 2012*; *Komori et al., 2014*). Most importantly, we never detected supernumerary neuroblast formation in the

btd, brat double type II neuroblast clone, indicating that the direct progeny of the btd mutant type II neuroblast were insensitive to the loss of brat function (**Figure 4J**). These data led us to conclude that btd mutant type II neuroblasts generate non-functional Ase⁻ immature INPs that likely adopt an identity of GMCs normally found in the type I neuroblast lineage. Thus, we conclude that Trx most likely maintains the type II neuroblast functional identity through btd.

## Over-expression of btd is sufficient to trigger a type I neuroblast to generate INPs

Because btd is necessary for the maintenance of a type II neuroblast functional identity, we tested whether over-expression of btd is sufficient to induce a type II neuroblast functional identity in a type I neuroblast. We induced GFP-marked lineage clones derived from single type I neuroblasts mis-expressing a UAS-btd transgene and assessed the identities of cells in the clones by examining the expression of cell fate markers. In the control clones, type I neuroblasts maintained Ase expression and generated GMCs (**Figure 5A**). Eighteen percent of type I neuroblasts mis-expressing btd lost Ase

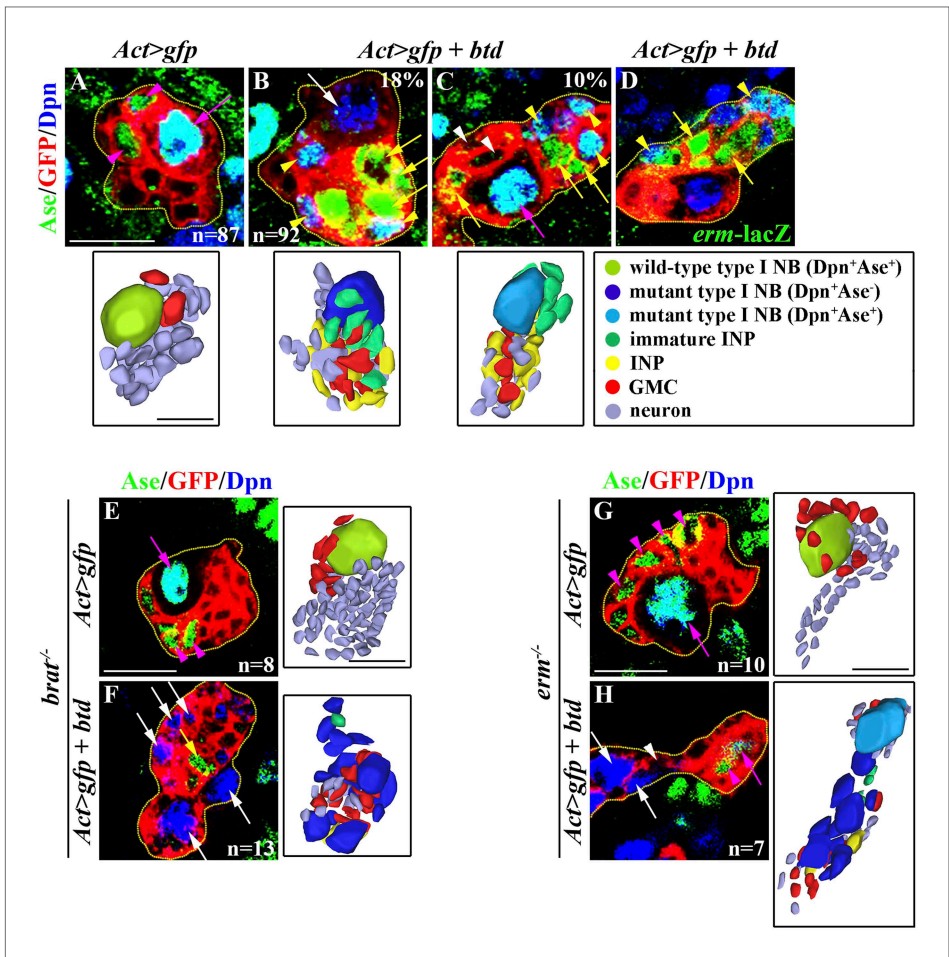

**Figure 5**. Over-expression of btd is sufficient to instruct a type II neuroblast functional identity in the type I neuroblast. (**A–E**) Over-expression of btd is sufficient to elicit a type II neuroblast functional identity. (**A–D**) In the 72-hr clones, 18% of type I neuroblasts over-expressing btd lose Ase expression and are surrounded by INP-like cells. An additional 10% of these neuroblasts maintain Ase expression despite being surrounded by INP-like cells. Three-dimensionally reconstructed images of the clones are shown to the right. Scale bar, 10 μm. (**E–H**) Progeny of type I neuroblasts over-expressing btd revert back to supernumerary neuroblast in the brat mutant or erm mutant. In the 72-hr clones, removing brat or erm function induces the formation of supernumerary type II neuroblasts derived from the progeny of type I neuroblasts over-expressing btd. Three-dimensionally reconstructed images of clones are shown to the right. Scale bar, 10 μm.

expression and generated progeny displaying a marker expression profile that is typically diagnostic of an immature INP or an INP (*Figure 5B,D*). Another 10% of type I neuroblasts mis-expressing *btd* generated progeny that resembled immature INPs or INPs by marker expression, but maintained Ase expression (*Figure 5C*). Thus, we conclude that mis-expression of *btd* is sufficient to trigger the characteristics that are specific for a type II neuroblast in a type I neuroblast.

We extended our analysis to assess whether mis-expression of *btd* might endow a type I neuroblast with the functional feature unique to a type II neuroblast—the competence to generate INPs. We reasoned that if a type I neuroblast mis-expressing *btd* indeed assumes a type II neuroblast functional identity, it should be able to generate immature INPs capable of maturing into an INP, a process critically dependent on the function of *brat* and *erm*. While removing *brat* function had no effects on the identities of progeny derived from control type I neuroblasts, it led to supernumerary type II neuroblast formation in the lineage clones derived from single type I neuroblasts mis-expressing *btd* (*Figure 5E–F*). Similarly, removing *erm* function also led to supernumerary type II neuroblast formation in the lineage clones derived from single type I neuroblast mis-expressing *btd* while not having any effects on the control type I neuroblast clones (*Figure 5G–H*). Since *brat* and *erm* function specifically in the immature INP to promote an INP identity (*Xiao et al., 2012*; *Janssens et al., 2014*; *Komori et al., 2014*), these data strongly suggest that mis-expression of *btd* was sufficient to endow a type I neuroblast with the competence to generate INPs. Thus, we conclude that *btd* plays an important role in eliciting the functional identity of a type II neuroblast.

## Btd mediates Trx-dependent maintenance of a type II neuroblast functional identity

Finally, we tested whether Trx maintains the type II neuroblast functional identity through *btd*. Consistent with our hypothesis, 40% of *trx* mutant type II over-expressing *btd* regained the characteristics that are specific for a type II neuroblast including loss of Ase expression and the generation of immature INPs and INPs (*Figure 6A–C*). Furthermore, over-expression of *btd* also significantly enabled *trx* mutant type II neuroblasts to generate INPs (*Figure 6D*). Thus, we conclude that *btd* is a key downstream target gene of Trx in the maintenance of the type II neuroblast functional identity.

## Discussion

Maintaining functionally distinct stem cell populations allows higher organisms to generate the requisite number of diverse cell types required for organogenesis. For example, neural stem cells in the subventricular zone and in the outer subventricular zone collectively contribute to the generation of all the cell types required for the development of a human brain (*Fietz et al., 2010*; *Hansen et al., 2010*). Similarly, heterogeneous stem cell pools have also been reported in other organs including the blood and intestine (*Barker et al., 2007*; *Graf and Stadtfeld, 2008*; *Copley et al., 2012*; *Marianes and Spradling, 2013*). Although the mechanisms that specify the identity of distinct stem cell types within a given organ have been proposed, the mechanisms that maintain the functional heterogeneity of stem cells have never been reported. In this study, we used the two well defined and functionally distinct types of neuroblasts in the fly larval brain to investigate the mechanisms that maintain stem cell functional heterogeneity during neurogenesis. We discovered that Trx functions uniquely to maintain a type II neuroblast identity through the H3K4 methylation activity of the SET1/MLL complex, thereby contributing to neuroblast heterogeneity during larval brain neurogenesis. We identified the homeodomain transcription factor Btd as a direct downstream target of Trx in the maintenance of a type II neuroblast identity. To our knowledge, this Trx-Btd-dependent mechanism provides the first mechanistic insight into the maintenance of stem cell functional heterogeneity within an organ (*Figure 7*). The homologs of Trx and Btd have been shown to play critical roles in regulating vertebrate neural stem cell functions (*Lim et al., 2009*; *MuhChyi et al., 2013*). Our findings lead us to speculate that the SET1/MLL histone methyltransferase complex might also contribute to the maintenance of stem cell heterogeneity in other higher eukaryotes.

## Trx maintains the type II neuroblast functional identity through the H3K4 methylation activity of the SET1/MLL complex

The SET1/MLL complex elicits biological responses by maintaining its target genes in an active state through the methylation of H3K4 (*Shilatifard, 2012*). Our data showed that the core components of the SET1/MLL complex is required for the maintenance of the H3K4 methylation in a type II neuroblast and the maintenance of a type II neuroblast functional identity (*Figure 3C–D,F*,

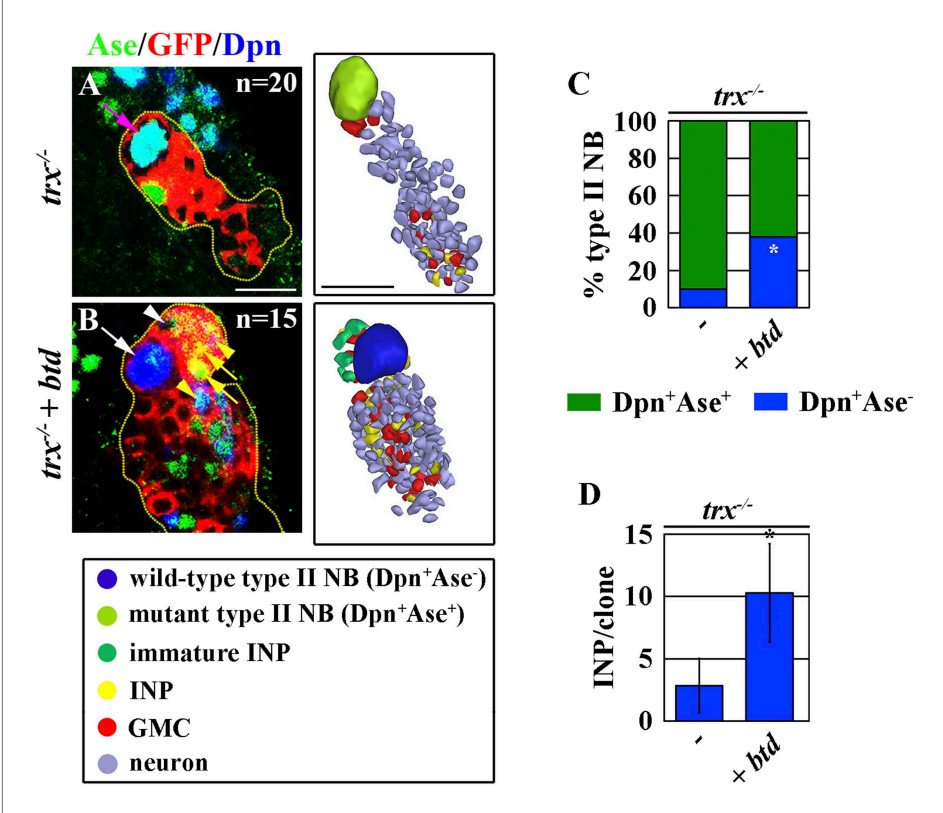

**Figure 6**. Over-expression of *btd* restores a type II neuroblast functional identity in *trx* mutant type II neuroblasts. (**A**–**D**) Overexpression of *btd* reinstates the ability to generate INPs in *trx* mutant type II neuroblasts. (**A**–**B**) In the 72-hr clones, while the control *trx* mutant type II neuroblasts are surrounded by GMCs, *trx* mutant type II neuroblasts over-expressing *btd* are surrounded by INP progeny. Three-dimensionally reconstructed images of the clones are shown to the right. Scale bar, 10 μm. (**C**) The neuroblast marker expression profile displayed by type II neuroblasts of the indicated genotypes. (**D**) The average number of INPs per clone of the indicated genotypes.

*Figure 3—figure supplement1*). Most importantly, over-expression of *rbbp5^FL^*, but not *rbbp5^SG^*, which encodes a mutant Rbbp5 protein that partially compromises the H3K4 methylation activity of the SET1/MLL complex (*Cao et al., 2010*), restored both H3K4 methylation and a type II neuroblast functional identity in *rbbp5* null type II neuroblasts (*Figure 3C–K*). These results indicate that the H3K4 methylation activity of the SET1/MLL complex is required for maintaining the functional identity of a type II neuroblast. In the fly genome, Trx, Trr, and dSet1 can each bind to the core components of the SET1/MLL complex (*Wu et al., 2008*; *Ardehali et al., 2011*; *Mohan et al., 2011*). Although the methylation activity of Trx was required for maintaining the type II neuroblast functional identity, removing *trx* function did not alter the global H3K4 methylation (*Figure 3A–B*, *Figure 4—figure supplement 1K–N*). In contrast, knocking down the function of *trr* or *dset1* did not affect the maintenance of a type II neuroblast functional identity despite resulting in the global loss of H3K4 mono- or tri-methylation (*Figure 4—figure supplement 1A–J*). These data strongly suggest that Trx maintains a type II neuroblast functional identity by regulating H3K4 methylation in specific downstream target loci.

## The Trx-Btd mechanism regulates the functional identity of a type II neuroblast

The functional identity of a type II neuroblast is defined by the competence of a neuroblast to generate INPs (*Weng and Lee, 2011*; *Homem and Knoblich, 2012*; *Janssens and Lee, 2014*). Our data indicate Trx plays a central role in maintaining the functional identity of a type II neuroblast by promoting the expression of a small number of genes (*Figures 1 and 4A*). We identified the *btd* gene as a critical downstream target of Trx that is both necessary and sufficient for the regulation of the type II

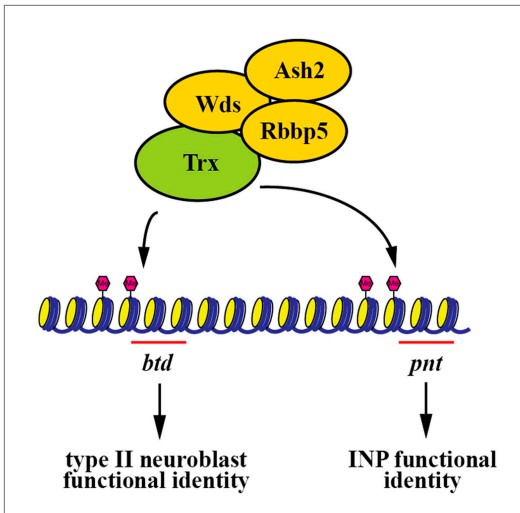

**Figure 7**. A summary model. The Trx histone methyltransferase complex maintains the type II neuroblast functional identity through the *btd* gene whereas it promotes INP identity specification through the *pnt* gene.

neuroblast functional identity (*Figures 4–7*). *btd* encodes a $C_2H_2$ zinc finger transcription factor required for proper patterning of the head segment during fly embryogenesis and likely functions as a transcription activator (*Wimmer et al., 1993*; *Schöck et al., 1999*). However, the role of Btd in regulating neuroblasts has never been established, and the mechanisms by which Btd elicits biological responses remain unclear. Several possible reasons exist to explain the relatively inefficient nature of eliciting the type II neuroblast functional identity in a type I neuroblast by the mis-expression of *btd* (*Figure 5*). First, certain co-factors might be required for Btd to efficiently activate its target gene transcription, and a lower abundance of these co-factors in type I neuroblasts hinders the functional output of mis-expressed Btd. Second, the epigenetic landscape might be vastly different between the two types of neuroblasts such that mis-expressed Btd may not have access to all of its target genes required to elicit the type II neuroblast functional identity in a type I neuroblast. Lastly, additional transcription factors might function in parallel with Btd to regulate the functional identity of a type II neuroblast. Btd is a highly conserved transcription factor (*Estella and Mann, 2010*; *MuhChyi et al., 2013*). Future studies to elucidate the mechanisms by which Btd regulates the functional identity of a type II neuroblast will provide critical insight in the regulation of neural stem cell heterogeneity during both invertebrate as well as vertebrate neurogenesis.

## The Trx-Pnt mechanism specifies an INP identity in the type II neuroblast lineage

We identified the *pnt* gene as another direct downstream target of Trx (*Figure 4A*, *Figure 4—figure supplement 2A*). We initially hypothesized that Pnt might function in parallel with Btd to maintain the functional identity of a type II neuroblast. This hypothesis was extremely appealing in light of a previous study demonstrating mis-expression of PntP1 can transform a type I neuroblast into a type II neuroblast (*Zhu et al., 2011*). Unexpectedly, knocking down the function of the *pnt* gene, which encodes at least three alternatively spliced transcripts, had no effect on the maintenance of the type II neuroblast functional identity, and instead, resulted in the formation of supernumerary type II neuroblasts (*Figure 4—figure supplement 2*). This result led us to revise our hypothesis and propose that Pnt functions in the immature INP to specify an INP identity. Consistently, heterozygosity of the *pnt* locus dominantly enhanced the supernumerary neuroblast in the *brat* or *erm* hypomorphic genetic background (*Figure 4—figure supplement 2G*). These two genetic backgrounds have been used extensively for elucidating the mechanisms that regulate the specification of an INP identity in the immature INP (*Xiao et al., 2012*; *Janssens et al., 2014*; *Komori et al., 2014*). Furthermore, over-expression of *pntP1* failed to restore the functional identity of a type II neuroblast in *trx* mutant type II neuroblasts (data not presented). Together, these data strongly suggest that *pnt* mainly functions to specify an INP identity rather than to maintain the type II neuroblast functional identity. Thus, we propose that in addition to maintaining the type II neuroblast functional identity, Trx also functions to promote INP identity specification through *pnt* (*Figure 7*).

## Attenuation of the competence to generate intermediate progenitor cells might provide a novel strategy to thwart the expansion of cancer stem cells

Strategies that uniquely target the functional properties of cancer stem cells will revolutionize cancer treatments. Cancer stem cells generate a hierarchy of progeny that include cell types directly contributing to the exponential expansion of cancer stem cells (*Magee et al., 2012*). Thus, reprogramming

their functional identity to bypass the cell types that directly contribute to the exponential expansion of cancer stem cells should halt further tumor growth. In our study, removing *trx* function efficiently reduced the number of supernumerary type II neuroblasts, which are proposed to serve as cancer stem cells in several *Drosophila* brain tumor models (*Caussinus and Gonzalez, 2005*; *Xiao et al., 2012*; *Eroglu et al., 2014*; *Janssens et al., 2014*; *Koe et al., 2014*; *Komori et al., 2014*), and increased the number of differentiated cells in the *brat* or *erm* mutant brain (*Figure 2*). Similarly, attenuating the competence of type II neuroblasts to generate INPs by removing *btd* function also efficiently halted the expansion of *brat* or *erm* mutant brain tumors (*Figure 4I–J*, data not presented). Our results strongly support the hypothesis that reprogramming the functional identity of putative cancer stem cells can significantly alter the course of tumorigenesis. As such, understanding the mechanisms that maintain stem cell heterogeneity during normal development might provide novel insight into designing rational therapies to promote switching of cancer stem cells to an alternative, non-cancerous stem cell type.

## Materials and methods

### Fly genetics and transgenes

Fly strains used in this study include Oregon R, *Ase-Gal4* (*Zhu et al., 2006*), *Ase-Gal80* (*Neumüller et al., 2011*), *brat*[DG19310], *brat*[k06028] and *brat*[11] (*Komori et al., 2014*), *erm*[1] and *erm*[2] (*Weng et al., 2010*), *erm-flag* (*Janssens et al., 2014*), *erm-lacZ* and *UAS-aPKC*[CAAX] (*Haenfler et al., 2012*), *pnt*[Δ88] (*Morimoto et al., 1996*), *trx*[Z11] (*Tie et al., 2014*), and *Wor-Gal4* (*Lee et al., 2006*). The following stocks were obtained from the Bloomington *Drosophila* Stock Center: *Elav-GAL4*, *Act-FRT-Stop-FRT-GAL4*, *ash2*[1], *btd*[XA], *FRT19A*, *FRT2A*, *FRT82B*, *GMR85C07-GAL4 (Btd-GAL4)*, *hs-flp*, *P(EP)G4226*, *pros*[17], *UAS-pnt*[RNAi] (TRiP.JF02227), *UAS-pnt*[RNAi] (TRiP.HMS01452), *trx*[E2], *tubP-Gal80*, *tubP-Gal80*[ts], *UAS-Dcr-2.D*, *UAS-mCD8-GFP*, and *UAS-trr*[RNAi] (TRiP.JF03242). We obtained the following stocks from the Vienna *Drosophila* RNAi Center *UAS-ash2*[RNAi] (100718), *UAS-dSet1*[RNAi] (40683), *UAS-pnt*[RNAi] (7171), *UAS-rbbp5*[RNAi] (106139), *UAS-trx*[RNAi] (108122), and *UAS-wds*[RNAi] (105371). *UAS-HA-btd*, *UAS-HA-pntP1*, *UAS-rbbp5*[FL]-*myc*, and *UAS-rbbp5*[SG]-*myc* were generated in this study by cloning the cDNA cloned into *p{UAST}attB* vector. The transgenic fly lines were generated via φC31 integrase-mediated transgenesis (*Bischof and Basler, 2008*). The *rbbp5* null allele was generated by imprecisely excising the *P(EP)G4226* element.

### Clonal analyses

Clones were induced following previously published methods (*Janssens et al., 2014*). Three-dimensional model of clones was generated using the *Mimics* software from *Materialize*, Leuven, Belgium. Confocal images were acquired using a Z-step size of 1.5 μm, and the identity of every cell within a clone was determined individually.

### Immunofluorescent staining and antibodies

Larvae brains were dissected in Schneider's medium (Sigma, St. Louis, MO) and fixed in 100 mM Pipes (pH 6.9), 1 mM EGTA, 0.3% Triton X-100, and 1 mM $MgSO_4$ containing 4% formaldehyde for 23 min. Larval brains were processed for immunofluorescent staining according to a previously published protocol (*Weng et al., 2012*). Antibodies used in this study include chicken anti-GFP (1:2000; Aves Labs, Tigard, OR), guinea pig anti-Ase (1:1000; Wang H), mouse anti-cMyc (1:100 Roche, Basel, Switzerland), mouse anti-Pros (MR1A; 1:500; DSHB, Iowa city, IA), rabbit anti-Ase (1:400), rabbit anti-β-gal (1:1000; MP Biomedicals, Santa Ana, CA), rabbit anti-H3K4me1 (1:500; Abcam, Cambridge, United Kingdom), rabbit anti-H3K4me3 (1:500; Active motif, Carlsbad, CA), rabbit anti-Phospho-Histone-H3(Ser10) (1:1000; EMD Millipore, Billerica, MA), rabbit anti-PntP1 (1:600; Skeath JB), rat anti-Dpn (1:2), rat anti-Mira (1:500). Secondary antibodies were from Jackson ImmunoResearch Inc., West Grove, PA. The confocal images were acquired on a Leica SP5 scanning confocal microscope (Leica Microsystems Inc., Buffalo Grove, IL).

### Chromatin immunoprecipitation

To obtain more than $2 \times 10^6$ supernumerary type II neuroblasts, we dissected 100 brains from *brat* mutant larvae aged for 4 days at 33°C in Schneider's medium (Sigma, St. Louis, MO) and fixed in 1.8% formaldehyde solution for 20 min. We stopped fixation by incubating the lysate with Glycine (0.25 M) at room temperature for 4 min and on ice for 10 min. Following fixation, samples were washed with wash buffer (1xPBS, 5 mM Tris–HCl pH7.5, 1 mM EDTA) containing proteinase inhibitors (Roche,

Basel, Switzerland) and 1 mM PMSF for three times and homogenized in SDS lysis buffer (1% SDS, 50 mM Tris–HCl pH8.1, 10 mM EDTA) to obtain nuclear extracts. The nuclear extracts were disrupted by using a sonicator (18 cycles of sonicating for 30 s and interval for 30 s). Five percent of the sonicated sample was stored for INPUT. The rest of the sonicated chromatin was incubated with antibodies in ChIP dilution buffer (0.01% SDS, 1.1% Trition X-100, 1.2 mM EDTA, 16.7 mM Tris–HCl pH8.1, 167 mM NaCl) at 4°C overnight. Samples were incubated with Dynal beads (Life technologies, Grand Island, NY) at 4°C overnight, washed twice with low salt immune complex wash buffer (0.1% SDS, 1% TritonX-100, 2 mM EDTA, 20 mM Tris–HCl pH8.1, 150 mM NaCl), twice with high salt immune complex wash buffer (0.1% SDS, 1% TritonX-100, 2 mM EDTA, 20 mM Tris–HCl pH8.1, 500 mM NaCl), three times with LiCl immune complex wash buffer (0.25 M LiCl, 1% NP40, 1% deoxycholate, 1 mM EDTA, 10 mM Tris–HCl pH8.1), twice with TE buffer, and then were eluted from beads. Cross-linking of chromatin–protein complex was reverted at 65°C overnight. Samples were treated with RNase A at 55°C for 2 hr and incubated with 2 µg of proteinase K at 45°C for 1 hr. Samples were cleaned up by phenol:chloroform and precipitated by EtOH precipitation. Samples were resuspended in 100 µl of water. 5 µl were used in each qPCR reaction. Antibodies used in this experiment were anti-Trx antibody (Mazo A), anti-H3K4me2 (07–030; Millipre, Billerica, MA), and rabbit IgG (ab46540; Abcam). The following individual specific primer sets were used for quantitative PCR: btd-E1, 5′-gttggccatt gcgtgtcctgtttc-3′ and 5′-gccccgctgcgctctatcca-3′, btd-E2, 5′-ggattaccgcagacgat-3′ and 5′-ggttggcc ggtggttgagt-3′, btd-TSS, 5′-cagcagcagcagcagcaacagt-3′ and 5′-gtcggcccgggtccaagtaa-3′, negative control, 5′-cagcagcagcagcagcaacagt-3′ and 5′-gtcggcccgggtccaagtaa-3′, pntP1-TSS, 5′-tttggtgttgttg tttttcttctt,-3′ and 5′-acgcgttctgttctgtttt-3′. Another negative control primer set was used in previously published paper (*Petruk et al., 2012*).

## qRT-PCR

Total RNA was extracted following the standard Trizol RNA isolation protocol (Life technologies, Grand Island, NY) and cleaned by the RNeasy kit (Qiagen, Venlo, Netherlands). First strand cDNA was synthesized from the extracted total RNA using First Strand cDNA Synthesis Kit for RT-PCR (AMV) (Roche, Basel, Switzerland). qPCR was performed using ABsolute QPCR SYBR Green ROX Mix (Thermo Fisher Scientific Inc., Waltham, MA). Data were analyzed by the comparative CT method, and the relative mRNA expression is presented. The following individual specific primer sets were used for quantitative PCR: *ase*, 5′-agcccgtgagcttctacgac-3′ and 5′-gcatcgatcatgctctcgtc-3′, *btd*, 5′-gcacggacgtacgcacaccaat-3′ and 5′-cctcggcggccaataccttct-3′, *dpn*, 5′-catcatgccgaacacaggtt-3′ and 5′-gaagattggccggaactgag-3′, *elav*, 5′-gcggcgcgtatcccattttcatct-3′ and 5′-tggccgcctcatcgtagttggtca-3′, *pntP1*, 5′-ggcagtacgggcagcaccac-3′ and 5′-ctcaacgcccccaccagatt-3′.

## Acknowledgements

We thank Drs Q Doe, M Dominguez, PJ Harte, R Mann, A Mazo, J Skeath, and H Wang for fly stocks and antibody reagents. We thank the Bloomington *Drosophila* Stock Center, Kyoto stock center, Vienna *Drosophila* RNAi Center and the Developmental Studies Hybridoma Bank for fly stocks and antibodies. We thank the BestGene Inc. for generating transgenic fly lines. We thank J Xu for technical advice on the chromatin immunoprecipitation experiment, and K Golden for proofreading the manuscript. We thank the members of the Lee lab for their intellectual input during the course of this study. HK was supported by a fellowship from the Japan Society for the Promotion of Science. DHJ was supported by a Cellular and Molecular Biology training grant (T32-GM007315). C-YL is supported by an NIH grant R01-GM092818.

## Additional information

### Funding

| Funder | Grant reference number | Author |
| --- | --- | --- |
| National Institute of General Medical Sciences | R01-GM092818 | Hideyuki Komori, Qi Xiao, Derek H Janssens, Cheng-Yu Lee |

The funder had no role in study design, data collection and interpretation, or the decision to submit the work for publication.

## Author contributions
HK, Conception and design, Acquisition of data, Analysis and interpretation of data, Drafting or revising the article, Contributed unpublished essential data or reagents; QX, Conception and design, Acquisition of data, Analysis and interpretation of data, Drafting or revising the article; DHJ, YD, Acquisition of data, Analysis and interpretation of data, Drafting or revising the article; C-YL, Conception and design, Analysis and interpretation of data, Drafting or revising the article

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
