## [Decision Letter]

Thank you for sending your work entitled “The Trithorax histone methyltransferase complex maintains the functional heterogeneity of neural stem cells” for consideration at *eLife*. Your article has been favorably evaluated by Fiona Watt (Senior editor), Marianne Bronner (Reviewing editor), and 2 reviewers, one of whom, Heinrich Reichert, has agreed to reveal his identity.

The Reviewing editor and the reviewers discussed their comments before we reached this decision, and the Reviewing editor has assembled the following comments to help you prepare a revised submission.

This manuscript reports the role of Trithorax (Trx) in the maintenance of type II neuroblast identity. Type II neuroblasts mutant for Trx gradually convert to type I neuroblasts: they turn on Asense and lose PntP1, and they generate GMCs instead of intermediate neural progenitors (INPs). Neuroblasts mutant for a core component of the SET1/MLL complex also show the same phenotype. By sequencing brains enriched for type I or type II neuroblasts, the authors identify a transcription factor, buttonhead (btd) whose transcripts are up-regulated in mutant brains enriched with type II neuroblasts. They show by ChIP that Trx binds to the *btd* gene. Type II neuroblasts mutant for *btd* do not turn on Ase, but their INPs never mature. Further, the authors show that overexpression of *btd* in type I neuroblasts is sufficient to convert them into type II neuroblasts that generate INPs (although this phenotype is only observed in ∼28% of neuroblasts). Finally, they show that over-expressing *btd* in type II neuroblasts that are mutant for trx rescues type II neuroblast identity. These findings are very interesting because they show that Trx maintains type II neuroblast identity through a specific transcription factor, *btd*, and thus add to our knowledge of the molecular control of type II neuroblasts identity in the fly, which are emerging as a good model for comparable neural stem cell in vertebrates. The experiments are performed appropriately and the results are presented in a largely logical manner. The data are generally solid and well-quantified. However, there are several significant points that need to be addressed:

1) The discovery of the contribution of *btd* to Type II neuroblasts appears to be more important than that of Trx and it is suggested to mention *btd* in the title.

2) One important missing point is that the author do not show the expression of *btd* in type II neuroblasts through antibody staining, or the loss of *btd* in Trx mutants. Although an enhancer fragment of *btd* drives Gal4 in type II neuroblasts, this is only a good indication that *btd* is in the right place. It would be better to see protein expression by antibody staining, if available.

3) *btd* mutant type II neuroblasts do not have the same phenotype as Trx mutants, suggesting that other target genes of Trx are involved. Could it be PntP1? Although *pntp1* mutants by themselves do not affect type II neuroblast identity, double mutant for *btd* and *pntp1* might have more severe phenotype than *btd* or *pntp1* mutants alone.

4) The reviewers strongly recommend deleting the discussion section on “Trx maintains a type II neuroblast functional identity independently of the Polycomb protein”. There is no mention or investigation of this issue at all in the Results section. Indeed in the Discussion section, the reader is confronted for the first time with largely undocumented data to support the section's conclusions. This is not acceptable, moreover it detracts from the quality of the experiments that are well described and documented in the Results section.

---

## [Author Response]

*1) The discovery of the contribution of* btd *to Type II neuroblasts appears to be more important than that of Trx and it is suggested to mention* btd *in the title*.

We have modified the title to “The Trithorax histone methyltransferase complex maintains the functional heterogeneity of neural stem cells through the transcription factor Buttonhead.”

*2) One important missing point is that the author do not show the expression of* btd *in type II neuroblasts through antibody staining, or the loss of* btd *in Trx mutants. Although an enhancer fragment of* btd *drives Gal4 in type II neuroblasts, this is only a good indication that* btd *is in the right place. It would be better to see protein expression by antibody staining, if available*.

Due to the lack of reagents, we were unable to perform the exact experiment that the reviewer suggested. We were unable to generate a specific antibody against the Btd protein for the immunofluorescent application. In addition, we were unable to recover any viable transgenic fly lines that carry a transgene containing a BAC clone covering the entire *btd* locus after two attempts. As the only option available, we examined the expression pattern of the *btd-Gal4* driver in the *rbbp5null* mutant brain. We observed a dramatic reduction in the expression of *btd-Gal4* in the *rbbp5null* mutant brain as indicated by the *UAS* reporter transgene expression. These data are consistent with our hypothesis that *btd* is a downstream target of Trx. We have now revised the text to include this result and added the image in Figure 4.

*3) Btd mutant type II neuroblasts do not have the same phenotype as Trx mutants, suggesting that other target genes of Trx are involved. Could it be PntP1? Although* pntp1 *mutants by themselves do not affect type II neuroblast identity, double mutant for* btd *and* pntp1 *might have more severe phenotype than* btd *or* pntp1 *mutants alone*.

We were initially extremely enthusiastic about the possibility that *pnt* acts in parallel with *btd* to maintain the functional identity of a type II neuroblasts because Trx binds to the promoter region of the *pntP1* transcription unit, and the expression of PntP1 was undetectable in *trx* mutant type II neuroblasts. Since the *pnt* locus encodes multiple alternatively spliced transcripts, we knocked down the function of all *pnt* isoforms in type II neuroblasts by over-expressing three distinct *UAS-RNAi* transgenes targeting the common exon. To our surprise, *pnt* mutant type II neuroblasts give rise to immature INP progeny that revert into supernumerary type II neuroblasts. This result strongly suggested that *pnt* functions in the immature INP to specify an INP identity. Consistently, the heterozygosity of *pnt* strongly enhanced the supernumerary neuroblast phenotype induced by genes required for proper specification of an INP identity, including *brat* and *erm*. Together, these data indicate that *pnt* functions in the immature INP rather than in the type II neuroblast. We attempted to knock down the function of both *btd* and *pnt* by over-expressing all possible combinations of available *UAS-btdRNAi* and *UAS-pntRNAi* transgenes in type II neuroblasts. We validated the effectiveness of the *btdRNAi* construct by inducing type II neuroblasts to assume the functional identity of a type I neuroblast in the larval brain. Larval brains co-expressing the *UAS-btdRNAi* and *UASpntRNAi* transgenes in type II neuroblasts possessed supernumerary neuroblasts that displayed a type II neuroblast marker expression profile. However, we were unable to interpret the identity of the supernumerary neuroblasts in the *btd*, *pnt* double mutant brain because *btd* mutant type II neuroblasts acquire a type I neuroblast functional identity without displaying a type I neuroblast marker expression profile. We have now revised the text and added a figure (Figure 4—figure supplement 2) to describe the functional characterization of *pnt* in type II neuroblasts.

*4) The reviewers strongly recommend deleting the discussion section on “Trx maintains a type II neuroblast functional identity independently of the Polycomb protein”. There is no mention or investigation of this issue at all in the Results section. Indeed in the Discussion section, the reader is confronted for the first time with largely undocumented data to support the section's conclusions. This is not acceptable, moreover it detracts from the quality of the experiments that are well described and documented in the Results section*.

We have now deleted this paragraph from the Discussion section.